# Hot Isostatic Pressing Control of Tungsten-Based Composites

Ryan Schoell [1], Aspen Reyes [1], Guddi Suman [1], Mila Nhu Lam [1], Justin Hamil [2], Samantha G. Rosenberg [1], LaRico Treadwell [1], Khalid Hattar [1,3] and Eric Lang [2,*]

1 Sandia National Laboratories, Albuquerque, NM 87185, USA
2 Department of Nuclear Engineering, University of New Mexico, Albuquerque, NM 87131, USA
3 Department of Nuclear Engineering, University of Tennessee, Knoxville, TN 37996, USA
* Correspondence: ejlang2@unm.edu

**Abstract:** Metal-oxide composites are commonly used in high temperature environments for their thermal stability and high melting points. Commonly employed with refractory oxides or carbides such as ZrC and HfC, these materials may be improved with the use of a low density, high melting point ceramic such as $CeO_2$. In this work, the consolidation of W-$CeO_2$ metal matrix composites in the high $CeO_2$ concentration regime is explored. The $CeO_2$ concentrations of 50, 33, and 25 wt.%, the $CeO_2$ particle size from nanometer to micrometer, and various hot isostatic pressing temperatures are investigated. Decreasing the $CeO_2$ concentration is observed to increase the composite density and increase the Vickers hardness. The $CeO_2$ oxidation state is observed to be a combination of $Ce^{3+}$ and $Ce^{4+}$, which is hypothesized to contribute to the porosity of the composites. The hardness of the metal-oxide composite can be improved more than 2.5 times compared to pure W processed by the same route. This work offers processing guidelines for further consolation of oxide-doped W composites.

**Keywords:** tungsten; metal-oxide composites; hot isostatic pressing

## 1. Introduction

Tungsten and its composites are commonly used in high temperature environments due to their ablation resistance, high melting point, and thermal shock resistance. These applications include rocket nozzles and noses of hypersonic materials where materials can be subjected to ablation, thermal shock, and high temperatures (~2000 °C and 3000 °C, respectively) [1–4]. Tungsten has a high melting point of 3422 °C but has poor oxidation resistance (oxidizing at ~400 °C), limiting potential uses of pure W in high temperature applications [2]. W composites have been used for numerous high temperature applications, including potassium-doped W [5] and oxide-doped tungsten, including $Y_2O_3$ [6,7] and $La_2O_3$ [8,9], showing improvements in (1) high temperature stability, (2) high temperature strength, and (3) creep resistance, while limiting recrystallization when compared to pure W [10]. In addition, transition metal carbide-W composites have been studied, with W-ZrC composites [11] and W-HfC composites [12] both showing enhanced ablation resistance.

Mechanical alloying can be used to fabricate these materials for both terrestrial and space applications, the latter of which has increased risk of ablation also contributing to the alloy's degradation [13]. An additional benefit of mechanical alloying and synthesis of W composites is that the grain and precipitate size is controllable. Controlling the grain size through alloying or composite synthesis allows for controlling material properties. Various W composites have been shown to limit W grain size and growth, including $HfO_2$ [14], $Y_2O_3$ [7], ZrC [15], TiC [16] allowing for control of W composite properties during and following high temperature exposure.

Powder consolidation, via sintering, hot pressing, or spark plasma sintering, is a common method for fabrication of dense W composites. The mechanisms of consolidation of dense specimens via sintering include surface diffusion, lattice diffusion, vapor transport,

and boundary diffusion, resulting in particle adhesion, followed by necking, and porosity isolation to complete the sintering [17]. However, sintering of a near-fully dense W compact requires high temperatures, high pressures, long hold times, or more often a combination of all three. Prior work has examined the kinetics of oxide-doped tungsten and the effects of wetted and non-wetted particles [18,19]. The sinterability of W-ThO$_2$, La$_2$O$_3$, Y$_2$O$_3$ and ZrO$_2$ composites depend on the oxide type, content, and sintering environment [20,21]. L. Chen investigated W-CeO$_2$ and W-HfO$_2$ composites sintering kinetics [22] and observed that ceria additions inhibited sintering of W at the early stages of sintering, with densification W-CeO$_2$ controlled by grain boundary diffusion [23]. However, in these prior studies, ceria additions were 1 wt.% (2.67 vol.%).

In this work, the microstructures of hot isostatic pressed W-CeO$_2$ composites at W-CeO$_2$ ratios from 3:1 to 1:1 by mass is investigated to study the microstructural impacts of high-content CeO$_2$ additions. By systematically varying the CeO$_2$ fraction, CeO$_2$ particle size, and consolidation temperature, investigating the density, phase, chemical bonding, mechanical properties, and consolidated particle size, design space guidelines for future W composites are offered. During consolidation, the CeO$_2$ partially reduces into Ce$_2$O$_3$ resulting in residual oxygen contents in the compacts, leading to lower relative densities and a significant decrease in Vickers hardness at higher CeO$_2$ concentrations, yet finer CeO$_2$ particles densify further. This work opens the door for further W-CeO$_2$ composite fabrication via Hot Isostatic Press (HIP) processing of similar powder consolidation methods.

## 2. Results and Discussion

### 2.1. XRD

The PXRD analysis was completed to determine the phases in the sample pre- and post-HIP. A pre-HIP processed puck PXRD pattern is shown in Figure 1. The pre-HIP sample PXRD patterns exhibited two main products, body-centered cubic tungsten (PDF# 00-004-0806) and cubic fluorite CeO$_2$ phase (PDF# 00-043-1002). As for the post-HIP samples, the profile A (1350 °C) and profile B (1850 °C) samples PXRD pattern are similar. For all post-HIP samples, the PXRD pattern exhibited the body-centered cubic tungsten (PDF# 00-004-0806), as shown in the pre-HIP sample. However, for all the post-HIP samples, the CeO$_2$ peaks exhibited a shift in 2θ up field as shown in the inset of Figure 2. Upon further investigation, the HIP conditions chosen introduce oxygen vacancies in the embedded CeO$_2$ particles, as confirmed by the spectrum shift to CeO$_{1.8}$ (CeO$_{1.8}$—PDF #04-003-6949). This indicates that the HIP process induces oxygen vacancy and an oxidation state changed from +4 to a mixture of +4/+3.6.

### 2.2. XPS

XPS analysis of the pre-HIP samples shows that the Ce (3d) region is a combination of Ce(IV) oxide and Ce(III) oxide which has been fit and reported by a number of papers [24–26]. The pre-HIP samples contain 64.85% Ce(IV) oxide and 35.15% Ce(III) oxide, and the post-HIP samples contain 65.76% Ce(IV) oxide and 34.24% Ce(III) oxide. However, the peak shape between the two samples is drastically different indicating a reduction from Ce(IV) to Ce(III) due to the HIP process, as shown in Figure 2a,b. Figure 2a,c show the spectrum of the pre-HIP sample, while Figure 2b,d show the spectrum post- HIP. Additionally, the HIP process causes the W to continue to be oxidized as seen by the increase in oxidized W components in Figure 2d.

The XPS analysis corroborates the PXRD results, showing a shift in the Ce oxidation state in the HIP process.

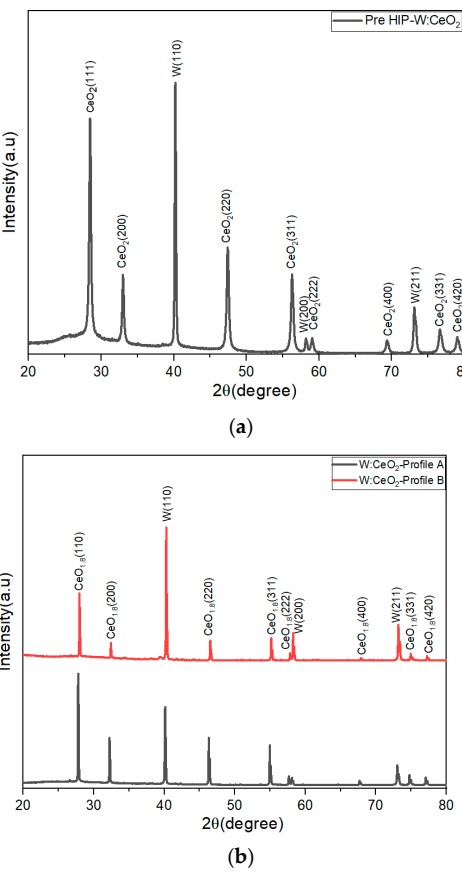

**Figure 1.** (**a**) PXRD analysis of 1:1 W-CeO₂ pre-HIP at temperature profile A and profile B conditions. (**b**) PXRD analysis of 1:1 W-CeO₂ composite and post-HIP at temperature under profile A and profile B conditions. The change from (**a**) to (**b**), shows the HIP process introduces a change in the Ce oxidation state, from only +4 before HIP to a combination of 4+/+3.6 post-HIP.

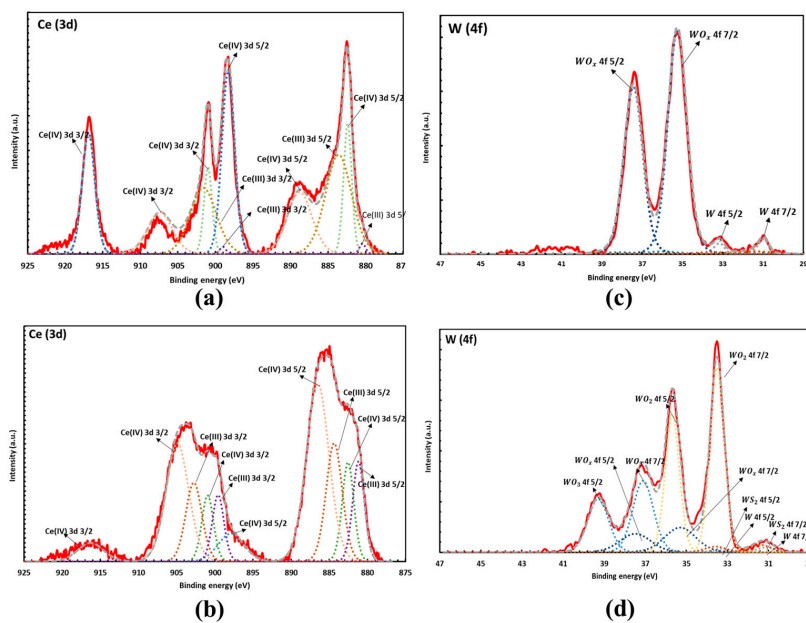

**Figure 2.** XPS spectra of pre-HIP sample in the Ce_3d region (**a**) and W_4f region (**c**). Post-HIP sample in the Ce_3d region (**b**) and W_4f region (**d**) showing the formation of mixed $Ce^{3+}$ and $Ce^{4+}$ after HIP. Peaks are labelled showing a combination of metallic and oxide peaks. The residual oxygen reacts with tungsten to form tungsten oxide during HIP.

### 2.3. Density

Density of post-HIP W-CeO$_2$ composites was determined by Archimedes density principle on multiples samples. In this method, the dry weight, suspended weight, and saturated weight of the W:CeO$_2$ sample of different ratios and sizes was determined. Figure 3 shows the densities for the composites for given function of CeO$_2$ concentration and size and HIP profile. The averaged density of HIP-processed W-CeO$_2$ samples at different HIP profiles are shown below in Table 1, with error representing one standard deviation of multiple tests on the same sample. The relative sintered density decreased with the increased CeO$_2$ content with less dependence on the size of CeO$_2$ particle. This may be due to increased residual oxygen content at high CeO$_2$ as the CeO$_2$ reduces, as shown with PXRD and XPS. However, it was observed that the density of the micro CeO$_2$ composites, regardless of HIP profile, is consistently higher than the nano CeO$_2$ composites fabricated with HIP profile A at all ratios. This trend indicates that the micro CeO$_2$ metal-oxide is a more homogenous composite, since nanoparticles tend to agglomerate more than micrometer size particles. At 1:1 ratio, less variation in the sintered density is observed across the HIP profiles and particle sizes, indicating difficulty in densifying high CeO$_2$ content composites, regardless of the HIP parameters, potentially due to the increase in oxygen content.

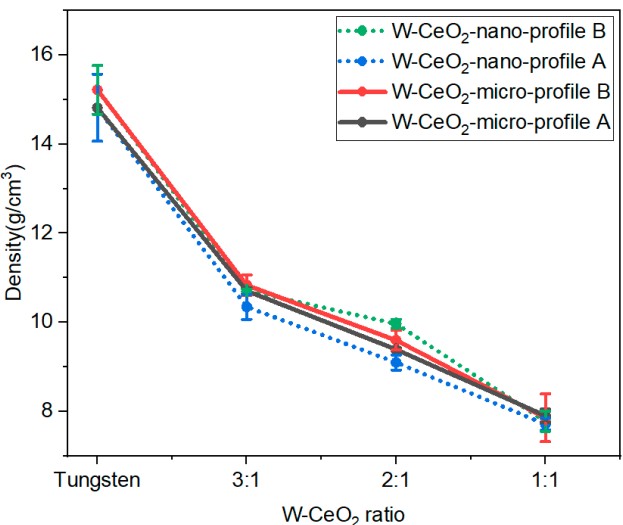

**Figure 3.** Density as a function of W-CeO$_2$ ratios showing an increase in CeO$_2$ content decreases the density. Sintering at higher temperatures enhances (Profile B) systematically increases the relative density regardless of starting CeO$_2$ particle size.

**Table 1.** HIP processing routes explored for the W-CeO$_2$ metal matrix composites and the resulting density, microstructure, and hardness.

| Sample Name | Maximum HIP Temperature | CeO$_2$ Powder Size | Density [g/cm$^3$] | Relative Density | W Particle Size [μm] | Vickers Hardness [HV] |
|---|---|---|---|---|---|---|
| 1:1 1–3 μm 1850 °C | 1850 °C | 1–3 μm | 8.617 ± 0.5374 | 0.64 | 7.6 | 223.7 ± 8.4 |
| 1:1 10–30 nm 1350 °C | 1350 °C | 10–30 nm | 7.714 ± 0.1322 | 0.58 | 2.3 | 224.6 ± 46.0 |
| 1:1 1–3 μm 1350 °C | 1350 °C | 1–3 μm | 7.902 ± 0.1513 | 0.59 | 5.4 | 155.1 ± 17.5 |
| 1:1 10–30 nm 1850 °C | 1850 °C | 10–30 nm | 8.091 ± 0.2227 | 0.61 | 3.7 | 138.42 ± 14.4 |
| 2:1 1–3 μm 1350 °C | 1350 °C | 1–3 μm | 9.392 ± 0.0368 | 0.62 | 3.9 | 230.1 ± 10.6 |
| 2:1 1–3 μm 1850 °C | 1850 °C | 1–3 μm | 9.598 ± 0.2199 | 0.63 | 4.2 | 280.2 ± 12.8 |
| 2:1 10–30 nm 1350 °C | 1350 °C | 10–30 nm | 9.100 ± 0.1697 | 0.60 | 2.4 | 230.0 ± 10.3 |
| 2:1 10–30 nm 1850 °C | 1850 °C | 10–30 nm | 9.966 ± 0.1004 | 0.66 | 3.7 | 300.0 ± 9.5 |

**Table 1.** *Cont.*

| Sample Name | Maximum HIP Temperature | CeO$_2$ Powder Size | Density [g/cm$^3$] | Relative Density | W Particle Size [μm] | Vickers Hardness [HV] |
|---|---|---|---|---|---|---|
| 3:1 1–3 μm 1350 °C | 1350 °C | 1–3 μm | 10.708 ± 0.0750 | 0.66 | 5.9 | 399.2 ± 57.9 |
| 3:1 1–3 μm 1850 °C | 1850 °C | 1–3 μm | 10.833 ± 0.2312 | 0.66 | 4.4 | 274.6 ± 3.0 |
| 3:1 10–30 nm 1350 °C | 1350 °C | 10–30 nm | 10.348 ± 0.2814 | 0.63 | 6.3 | 320.2 ± 27.4 |
| 3:1 10–30 nm 1850 °C | 1850 °C | 10–30 nm | 10.718 ± 0.0198 | 0.66 | 7.6 | 255.4 ± 6.2 |
| W | 1350 °C. | N/A | 14.81 ± 0.7531 | 0.76 | N/A | 147.5 ± 4.3 |
| W | 1850 °C | N/A | 15.19 ± 0.5494 | 0.79 | N/A | 129.7 ± 6.4 |

### 2.4. Microstructure

Figure 4 shows Backscatter Electron (BSE) and Secondary Electron (SE) SEM micrographs. Pores are evident in the high-magnification images in Figure 5. EDS compositional maps shown in Figure 4 show the fraction of the sample surfaces Ce-rich with pockets of W. Selected EDS maps are chosen for the CeO$_2$ compositions and HIP parameters. As the W content increases, from 1:1 W:CeO$_2$ to 3:1 W:CeO$_2$, an increase in surface fraction that is W is observed. The EDS maps show a correspondence between regions of Ce content and O content.

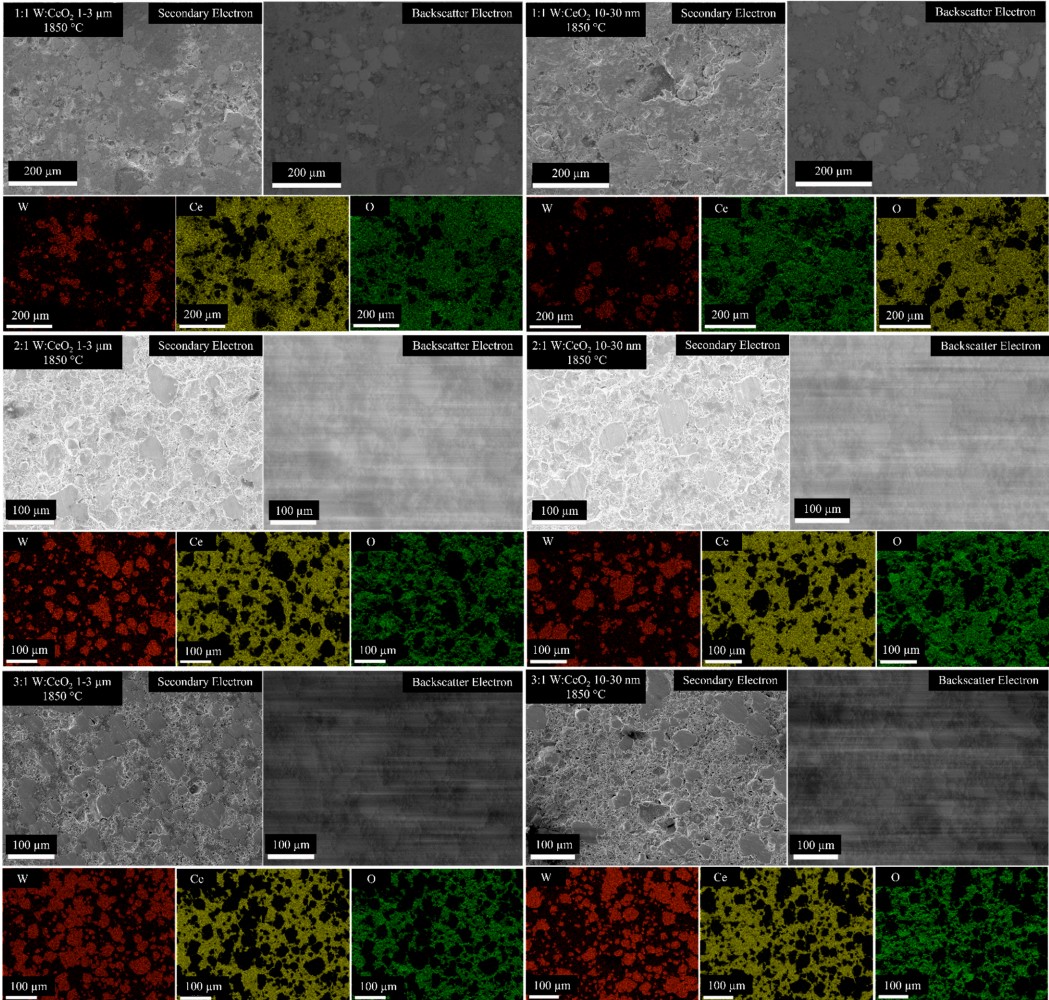

**Figure 4.** SEM micrographs and SEM-EDS compositional maps of W, Ce, and O of the several W:Ce composites processed by the conditions labeled.

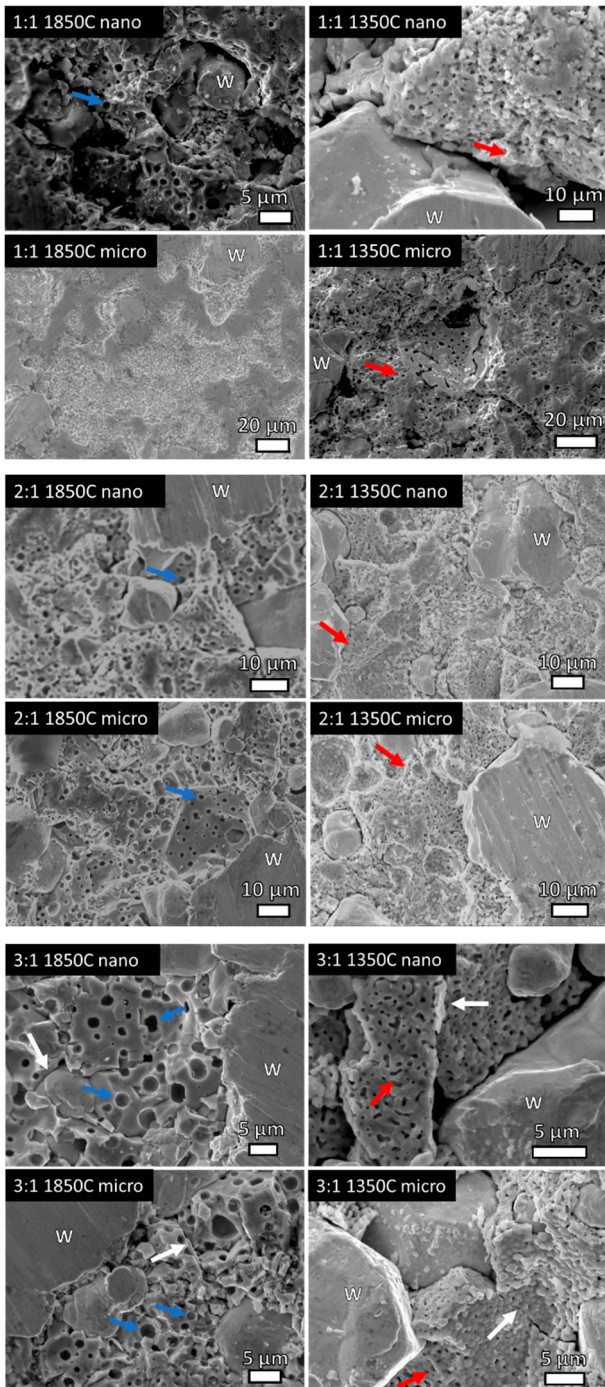

**Figure 5.** High-magnification SE SEM micrographs showing porosity of the CeO$_2$ regions in as-indicated specimens. HIP at 1850 °C shows larger, micron-sized pores and dodecahedron proto-CeO$_2$ grains indicative of greater consolidation. Red arrows indicate necking, white arrows indicate grain boundaries, and blue arrows indicate pores. W indicates locations of tungsten (W) grains.

High magnification SE micrographs (Figure 5) of select samples show pores present in the CeO$_2$-rich regions. At 2:1 and 3:1 compositions when HIP was performed at 1850 °C regardless of CeO$_2$ particle size, ~1 μm-sized pores within polyhedral-shaped CeO$_2$ grains are observed and indicated with blue arrows. This represents the start of sintering, and the closed porosity indicates further sintering. HIP at 1350 °C indicates more connected porosity and CeO$_2$ powder particles are identifiable, indicative of less consolidation and worse sinterability. Thus, further sintering is observed with the 1850 °C HIP and the

microstructures confirm the density studies where 1850 °C HIP resulted in higher densities than 1350 °C.

Labels have been added to indicate W regions, with non-labeled regions as $CeO_2$. White arrows indicate W-$CeO_2$ grain boundaries, red arrows indicate locations of $CeO_2$ particles necking, and blue arrows indicate locations of pores in the $CeO_2$ grains. In addition, Table 1 gives the average W particle size and Vickers hardness values for all consolidated specimens, along with the HIP parameters. The maximum densities for each composition are determined via rule of mixtures of densities of W and $CeO_2$, respectively. Figure 6 shows the Vickers hardness values as a function of sample composition for each HIP condition, with error representing one standard deviation of multiple measurements on each sample.

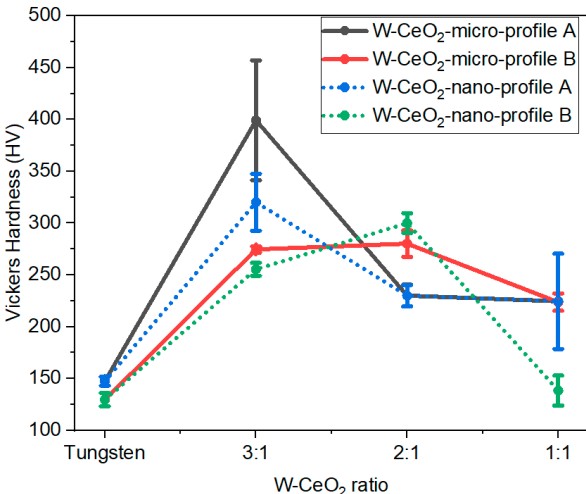

**Figure 6.** Vickers hardness values for all consolidated specimens. Increasing the $CeO_2$ content is observed to correlate with decreases in the hardness values.

A general trend in hardness is observed as increasing the W content from 1:1 to 3:1, until progressing to the pure W samples. These trends correlate with the relative density of the samples, indicating that control of the densification will allow for control of and improved mechanical properties. For the HIP processing conditions investigated, the addition of $CeO_2$ increases the hardness of the composites compared to pure W.

## 3. Discussion

Hot isostatic pressing control of the temperature and pressure offers fine control of the resulting microstructures. This work examines the interplay between particle size, particle concentration, and HIP parameters for W-$CeO_2$ composites. The reduction of the $CeO_2$ and the development of mixed $CeO_2$ and $Ce_2O_3$ influence the formation of the numerous pores on the Ce-rich regions of the composites. At the high temperature isostatic hold step, the production of oxygen gas corresponding with the production of $Ce_2O_3$ accumulates and the isolated oxygen gas bubbles prevent the complete densification of the composites, observed previously by Zhou et al. [27]. The pores observed on the surfaces of the $CeO_2$ compacts indicate a non-fully dense microstructure, with this form of pores indicative of the densification process halting prior to the final stages of sintering [17]. The $CeO_2$ powders HIP processed at 1350 °C show necking in Figure 6, indicative of Stage 1 sintering. Increasing the HIP temperature to 1850 °C, the development of dodecahedron-shaped $CeO_2$ grains with intergranular, isolated pores indicated the progression to Stage 2 sintering [17]. In prior work sintering Cu:$CeO_2$ composites, $CeO_2$ concentrations up to 20 wt.% showed finely dispersed Ce throughout Cu grains but showed intergranular cracks [28].

Sintering of pure $CeO_2$ has shown that decreasing the particle size increases densification rate sintering at both 1200 °C and 1400 °C, while $CeO_2$ consolidated with coarser

particles showed less grain growth under high temperature exposure [29]. Similarly, a study of $CeO_2$ by Zhou, et al. examined sintering above 1200 °C, observing maximum density at 1200 °C with decreasing density above that temperature, attributed to the formation of oxygen gas pores. Zhou also observed that micron-sized $CeO_2$ particles densified at higher temperatures compared to nano-sized powders, however the nano-sized powders showed more weight loss (off-gas of oxygen) at the same temperatures than the coarser powders, corresponding with greater surface area [27].

In this work, decreasing the $CeO_2$ particle size leads to lower overall densities, regardless of concentration and HIP temperature. Consequently, at the high $CeO_2$ concentrations studied here, the residual porosity present will likely decrease the mechanical strength, observed here in the lower Vickers hardness values. Generally, as the $W:CeO_2$ ratio increases, the hardness values increase, corresponding with an increase in the relative density. However, all $CeO_2$-doped W samples show higher hardness than the pure W sample. This indicates that there is a fine relationship between HIP parameters, composition, and resulting properties of W composites. The relative density of the W samples is ~77%, which is lower compared to the $CeO_2$ composites. Di Fonseca, et al. showed that the hardness of $Cu:CeO_2$ composites increased as the $CeO_2$ content increased from 8 to 20% [28]. Therefore, exploring the range of $W:CeO_2$ ratios more extensively (from the 1% to 25% levels) will offer more insights into the densification and hardness trends.

The W-rich regions in the composites studied here show high density, with no pores within W regions. However, the $W\text{-}CeO_2$ interfaces do not appear well formed with numerous pores and cracks preventing full interfacial bonding, a consequence of the low density of the samples. Previous work of sintering of pure W establishes that the HIP parameters here, for pure W, indicate Stage 2 sintering (volume diffusion) for the 1850 °C HIP and Stage 1 sintering (surface diffusion) for the 1350 °C HIP [17]. In the $W\text{-}CeO_2$ composites, the size of the constituent W particles is relatively constant across HIP parameters, $CeO_2$ particle sizes, and concentrations. The W rich-regions have similar sizes to that of the starting particle size, yet no W-to-W particle necking is observed, indicating that for the W particles in contact with each other full sintering occurred, yet for those not in contact with one another the $CeO_2$ prevented the formation of necking and limited densification. Thus, future work will explore higher temperature HIP processing to complete densification for W grains.

Previous work has shown that sintering of HfC-W composites leads to the formation of $HfO_2$ following ablation [30], as the HfC transforms to $HfO_2$ to protect the volatile and lower melting point $WO_3$. The work presented here explores fabrication of W composites directly with an oxide at 25 vol.%, 33 vol.%, and 50 vol.% $CeO_2$ additions to W, with the as-HIP produced a bonding state that is a combination of $CeO_2$ and $Ce_2O_3$. Precise control of the bonding state may allow for enhanced oxidation or ablation resistance, which may be achieved through control of the $CeO_2$ additions and processing parameters.

This work establishes the HIP parameters that best consolidate the desired microstructures of $W\text{-}CeO_2$ composites with high $W\text{-}CeO_2$ ratios. Expanding on prior work investigating similar concentrations of $CeO_2$ [22], ZrC, and HfC [11,12], this work explores a new space of $W\text{-}CeO_2$ composites, indicating how reducing the $CeO_2$ concentration and increasing the HIP temperature allow for finest control over the W grain size and increases the final density. In this work, HIP shows consolidation of W and $CeO_2$ composites by XRD with a transformation to $CeO_2$ and $Ce_2O_3$ phases. The reduction of Ce is consistent with loss of oxygen from the $CeO_2$ compound, which may result in free residual oxygen in the compact during HIP. Thus, this may result in the high density of pores present in the $CeO_2$ rich regions shown in Figure 6. Formation of gaseous oxygen in the HIP in a static environment may introduce more porosity during HIP. Nonetheless, the addition of $CeO_2$ increases the Vickers hardness significantly compared to pure W, showing that $CeO_2$ addition can improve the mechanical performance of HIP W. The highest hardness values at the 3:1 ratio indicates that the properties of W can be tailored through $CeO_2$ content

and low $CeO_2$ concentrations offer the best combination of properties and structure for future studies.

Future work to consolidate denser composites at higher temperatures will use a flowing gas environment to mitigate oxidation concerns. Future work will focus on the performance of these composites in extreme high temperature environments, including oxidizing and ablative environments, thermal transients, and determining the bulk mechanical properties in application-relevant environments. Additionally, future work will reduce the $CeO_2$ concentrations to ~1-2 vol.% to achieve a dispersoid microstructure with fine-grained W and a high density of small, dispersed second phase particles better suited for improving creep strength and radiation resistance. Also, sintering in flowing gas environments may be used in the future to limit the accumulation of oxygen gas in pores to hopefully increase density. This future work controlling HIP atmospheres and smaller $CeO_2$ concentrations will refine HIP processing necessary to produce tailored microstructures and subsequent properties.

## 4. Materials and Methods

### 4.1. Materials

Isopropyl alcohol (IPA) (ACS Reagent, ≥99.5%) was purchased from Sigma Aldrich. Commercial tungsten powder (10 μm, >99.99% trace metals, Tungsten carbide manufacturing), cerium oxide powder of size (1–3 μm, 99.9%) (micro ceria) and cerium oxide powder of size (10–30 nm) (nano ceria) were purchased from SkySpring Nanomaterials, Inc. All materials were of analytical grade and used as received.

### 4.2. HIP Processing

$W$-$CeO_2$ composites were prepared via a hot isostatic pressing (HIP) processing procedure of the respective W-metal and micro- and nanosize $CeO_2$. In a standard material processing method, the various ratios of starting materials were grounded in a mortar to insure a homogenous mixture. After mixing the materials, the mixtures were pressed into a pellet using a die set of 1 inch-diameter using a Carver press at 17,000 pounds. The pellet was then treated in a HIP for further reaction under profile A and profile B reactions conditions seen in Figure 7. Before profile A and profile B was conducted, the HIP was vacuumed ($5 \times 10^{-4}$ Torr) and argon purged (100 psi) for 3 cycles. All ratios and reactions conditions were conducted in argon atmosphere.

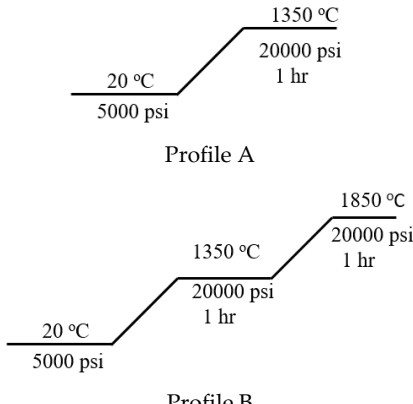

**Figure 7.** HIP temperature/pressure profiles A and B under argon gas with a ramp rate of 5 °C/min rate.

### 4.3. Characterization Methods

Analytical data were collected on solid pucks after HIP procedure. Powder X-ray Diffraction (PXRD) of the products were conducted on a zero-background holder by a Siemens D500 Diffractometer equipped with a graphite monochromatic with Cu Kα radiation (1.5418 Å) operating at 40 kV and 40 mA. The samples were scanned via X-rays at a rate of 4°/s in 2θ range from 20–80°. The PXRD patterns of samples were analyzed using

the JADE 9.6.0 software with ICDD PDF-4 database. Archimedes density experiments were conducted to calculate the density of all products. X-ray Photoelectron Spectroscopy (XPS), measurements were taken on a Kratos AXIS Supra, using a monochromatic Al K$\alpha$ X-ray source. Surveys and high resolution for each element were taken on each sample. The surveys had a pass energy of 160 eV, while the pass energy for the high-resolution elements was 20 eV. The data was analyzed with CASA XPS, by aligning the carbon spectra to 285 eV and using RSF for the Kratos system where F (1s) is set to 1. Scanning Electron Microscopy (SEM) including Energy Dispersive x-ray Spectroscopy (EDS) was performed using a JEOL IT 300 HR/LV SEM with EDAX Octane Elite Super EDS Detector. Backscatter electron and secondary electron SEM micrographs, as well as EDS compositional maps, were collected with 20 kV accelerating voltage. Specimens were mechanically polished with 400 grit SiC paper prior to imaging in the SEM. Analysis of the W particle sizes for each sample was performed in ImageJ [31]. Following SEM, Vickers hardness measurements were performed with an MMT-X series, MATSUZAWA microhardness tester on consolidated specimens using a 1000 g load with a 10 s dwell time. At least five indents were performed on each sample.

## 5. Conclusions

W-CeO$_2$ metal-oxide composites are investigated for use in high temperature, oxidizing environments for their high melting point and reduced density. In this study, the control of the microstructure through systematic variations of HIP temperature, CeO$_2$ particle size, and CeO$_2$ concentration is investigated. Control of the microstructure is dictated primarily by the HIP temperature, while decreasing CeO$_2$ particle size is an effective way of increasing the relative composite density. The reduction of CeO$_2$ into Ce$_2$O$_3$ during HIP processing changes the microstructure, forming tungsten oxide and gas bubbles that result in porosity and limits the densification of the composite. This initial investigation into HIP processing of metal-oxide composite of W-CeO$_2$ should both good microstructural control and an increased in hardness of over 2.5 times. This initial work into HIP processing of W:CeO$_2$ offers compositional and processing guidelines for future W composite materials.

**Author Contributions:** Conceptualization, L.T. and K.H.; Methodology, E.L.; Formal analysis, R.S., A.R., G.S., M.N.L., J.H., S.G.R. and E.L.; Investigation, R.S., A.R., G.S., M.N.L., S.G.R. and E.L.; Resources, K.H.; Data curation, J.H.; Writing—original draft, R.S. and E.L.; Writing—review & editing, A.R., G.S., M.N.L., S.G.R., L.T. and K.H.; Visualization, G.S., M.N.L., J.H. and E.L.; Supervision, E.L.; Funding acquisition, L.T. and K.H. All authors have read and agreed to the published version of the manuscript.

**Funding:** This work was performed, in part, at the Center for Integrated Nanotechnologies, an Office of Science User Facility operated for the U.S. Department of Energy (DOE) Office of Science. Sandia National Laboratories is a multimission laboratory managed and operated by National Technology & Engineering Solutions of Sandia, LLC, a wholly owned subsidiary of Honeywell International, Inc., for the U.S. DOE's National Nuclear Security Administration under contract DE-NA-0003525.

**Institutional Review Board Statement:** Not applicable.

**Informed Consent Statement:** Not applicable.

**Data Availability Statement:** The data presented in this study are available on request from the corresponding author.

**Acknowledgments:** The authors would like to thank Cody Corbin for helpful discussion. This work was performed, in part, at the Center for Integrated Nanotechnologies, an Office of Science User Facility operated for the U.S. Department of Energy (DOE) Office of Science. Sandia National Laboratories is a multimission laboratory managed and operated by National Technology & Engineering Solutions of Sandia, LLC, a wholly owned subsidiary of Honeywell International, Inc., for the U.S. DOE's National Nuclear Security Administration under contract DE-NA-0003525. The views expressed in the article do not necessarily represent the views of the U.S. DOE or the United States Government.

**Conflicts of Interest:** The authors declare no conflict of interest.

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
