# Peer review of "Hot Isostatic Pressing Control of Tungsten-Based Composites"

_inorganics, doi:10.3390/inorganics11020082_

Round 1

Reviewer 1 Report

In this paper, authors present the study on the HIP process of W-CeO2 composite, including the effects in oxide composition, powder size, and profile. The work is of good interest in the area of fabricating W composites with a variety of engineering applications. The writing of the manuscript is concise and well organized. A few details need to be checked and improved before the acceptance of the manuscript.

Authors mentioned that the density of micro CeO2 composites is consistently higher than nano CeO2 composites at all ratios (Line 154~156). However, this seems not clear in that reported in Table 1, which tells that in the 2:1 ratio case, the relative densities for micro composites are 0.62 and 0.63, while the relative densities for nano composites include a larger value of 0.66. The "consistently higher" needs more explanation. Also, in the 1:1 ratio case, the relative densities of micro composite in 1350C profile and 1850C profile are 0.59 and 0.64, while for nano composite in these two profiles are 0.58 and 0.61. The difference between nano composite densities is smaller. Then the conclusion of the micro composite has less dependence on HIP profiles needs more explanation.

Figure 3 has four images but only two labels and not positioned well. Consider using four labels, make better label positions, and provide more details in the caption of Figure 3.

The small XRD pattern figure at the top of Figure 2b, the labels and scale bars in Figure 5 are not clear. It will be better to remake them with higher fidelity. 

In Figure 6, the typical features like different types of grains, pores, connected pores, necking effects, etc. described in the text should be pointed out (such as by adding some arrows) in the SEM images to make it clearer to readers.

Reviewer 2 Report

In this work, Authors studied the influence of CeO2 concentration in W-CeO2 metal matrix composites. Although this is relevant work, the article lacks depth. In depth discussion is required with evidence and must support results with the literature. Also, the novelty statement must be clearly stated. Furthermore, the language is below the standard and should be thoroughly revised.

Following are the additional comments which require attention:

Figure 5 needs more discussion. Figure 6, Sem images need to be labeled to support the statement given in the discussion.

Vickers hardness tester details and discussion of result is missing.

How authors selected the range of CeO2 concentration. Why the concentration was not higher or lower than the selected concentration. The optimum concentration of the CeO2 should be discussed clearly.

Compression strength results must be added to increase the knowledge of the manuscript and support other mechanical tests (e.g., hardness and density results).

Properties mentioned in table 1 must be presented graphically for a quick comparison.

It is advisable to follow third person writing in academic writing, e.g. the frequent use of word “WE” can be avoided.

Round 2

Reviewer 2 Report

No comment.